

# Age stratigraphy in the East Antarctic Ice Sheet inferred from radio echo sounding horizons

Anna Winter[1], Daniel Steinhage[1], Timothy T. Creyts[2], Thomas Kleiner[1], and Olaf Eisen[1,3]

[1]Alfred-Wegener-Institut, Helmholtz-Zentrum für Polar- und Meeresforschung, Bremerhaven, Germany
[2]Lamont-Doherty Earth Observatory, New York, NY, USA
[3]Universität Bremen, Bremen, Germany

**Correspondence:** Olaf Eisen (olaf.eisen@awi.de)

**Abstract.** The East Antarctic Ice Sheet contains a wealth of information that can be extracted from its internal architecture such as distribution of age, past flow features and surface and basal properties. Airborne radar surveys can sample this stratigraphic archive across broad areas. Here, we identify and trace key horizons across several radar surveys to obtain the stratigraphic information. We transfer the age-depth scales from ice cores to intersecting radar data. We then propagate these age scales across the ice sheet using the high fidelity continuity of the radar horizons. In Dronning Maud Land, including Dome Fuji, we mapped isochrones with ages of 38 ka and 74 ka. In the central region of East Antarctica around Dome Concordia, Vostok, and Dome Argus, we use isochrone ages of 38 ka, 48 ka, 90 ka, and 161 ka. Taking together both regions, we provide isochrone depths traced along a combined profile length of more than 40,000 km and discuss uncertainties of the obtained stratigraphy, as well as factors important to consider for further expansion. This dataset the most extensive distribution of internal horizons in East Antarctica to date. The isochrone depths are available on PANGAEA: https://doi.pangaea.de/10.1594/PANGAEA. 895528.

## 1 Introduction

The internal stratigraphy of the East Antarctic Ice Sheet (EAIS) provides information about both the climate archive contained within the ice as well as the dynamic history of the ice sheet. The stratigraphic information is an important complement to ice-core analyses. It can improve the interpretation of rate and magnitude of climate changes from ice-core records by informing about the flow path and provenance of core ice and irregularities that potentially disturb the age stratigraphy at the core site (e.g., Fahnestock et al., 2001; NEEM community members, 2013; Parrenin et al., 2004). With a direct connection of different deep ice cores by continuous stratigraphic horizons, the ice core age scales can be synchronized and the uncertainties reduced (Cavitte et al., 2016; MacGregor et al., 2015a; Steinhage et al., 2013). With additional information or assumptions about the thinning of deposited snow layers, information about past accumulation rates and their spatial and temporal variation can be inferred away from available ice-core locations (e.g., Cavitte et al., 2017; Huybrechts et al., 2009; Koutnik et al., 2016; Leysinger Vieli et al., 2011; MacGregor et al., 2009; Neumann et al., 2008; Waddington et al., 2007). The information about past accumulation rates constrains the age resolution at potential future ice-core drill sites. Furthermore, an established stratigraphy ensures an undisturbed layering and facilitates age extrapolation of the maximum expected age at the coring





location (Fischer et al., 2013). These uses of stratigraphic information for future drill-site selection are largely exploited in the currently running reconnaissance studies for the International Partnership in Ice Core Sciences (IPICS) Oldest Ice target, aiming at the retrieval of a continuous 1.5 Ma old ice core record (Brook et al., 2006; Karlsson et al., 2018; Parrenin et al., 2017; Van Liefferinge and Pattyn, 2013; Van Liefferinge et al., 2018; Young et al., 2017).

Information about the ice sheet dynamic history that is contained in its stratigraphy allows us to understand past behaviors of the ice sheet and asses predictions about future ice sheet behavior (e.g., Cook et al., 2013; Ritz et al., 2015; Gulick et al., 2017). The internal stratigraphy responds to ice sheet dynamics and integrates changes in flow in the location and shape of the horizons. Movement of dome positions and ice divides will modify the horizons. Similarly, migration of ice streams can locally enhance divergence that pulls down stratigraphy (e.g., Bingham et al., 2007; Clarke et al., 2000; Conway et al., 2002;

Jacobel et al., 2000; Raymond, 1983; Siegert et al., 1998b; Urbini et al., 2008; Whillans, 1976).

Once established, an accurate age-depth stratigraphy can provide useful constraints for the evaluation of those ice-flow models which incorporate age tracers (e.g., Hindmarsh et al., 2009; Parrenin et al., 2017; Sutter et al., 2015, 2018). Furthermore, the combination of the stratigraphy with models adjusted to the specific application or region can provide additional information about past ice-sheet dynamics.

Radio-echo sounding (RES) is the method of choice to establish an age stratigraphy over broad areas in ice sheets. RES measurements reveal deep internal reflection horizons (IRHs), that mainly originate from contrasts in conductivity. The conductivity signals (e.g. acid from volcanic eruptions) are deposited from the atmosphere and thus form isochronous horizons (Eisen et al., 2006; Fujita et al., 1999; Gogineni et al., 1998; Jacobel and Hodge, 1995; Millar, 1981; Paren and Robin, 1975; Siegert et al., 1998b). We assign ages to isochronous IRHs from the age-depth scales of deep ice cores where the RES profiles

pass close the ice-core drill sites.

The age stratigraphy of the EAIS has only been obtained for several portions (e.g., Siegert et al., 1998b, a; Steinhage et al., 2001; Leysinger Vieli et al., 2011; Cavitte et al., 2016) and is thus incomplete. Roadblocks in achieving a whole-continent stratigraphy include the large amount of time required to trace internal horizons, the incomplete spatial coverage of data, and only few studies investigating how stratigraphic information from different RES systems can be integrated (Cavitte

et al., 2016; Winter et al., 2017). The many applications of a dated RES stratigraphy have motivated the SCAR Action Group AntArchitecture, designed to overcome the stated difficulties and establish such a stratigraphy for the complete Antarctic Ice Sheet (Bingham, pers. comm., 2018). An established stratigraphy can be used to better understand the ice sheet, as has been demonstrated for the Greenland example by MacGregor et al. (2015a, b, 2016a, b).

Our study contributes to establishing the internal horizon architecture, how it varies over long distances, and how the age-

depth relationship changes. We use airborne RES data that have been collected by the Alfred Wegener Institute (AWI) in numerous campaigns during the last 20 years (Steinhage et al., 2001, 2013), and RES data collected across the Gamburtsev subglacial mountains with tie lines to other surveys (e.g., Bell et al., 2011; Das et al., 2013). The IRHs are assigned with their ages from the age scales of the Dome Fuji ice core (Dome Fuji Ice Core Project Members: Kawamura et al., 2017) and the ice cores in Dronning Maud Land (EDML, Oerter et al., 2004; Ruth et al., 2007) and at Dome Concordia (EDC, EPICA

community members, 2004). We track these IRHs along 40,000 km of RES profiles across the ice sheet.



## 2  Data and Methods

### 2.1  RES data

We use RES data that were collected by the AWI between 1998 and 2008 and, additionally, the AGAP-South data over the Gamburtsev Mountain Province in the center of East Antarctica. The systems' characteristics are summarized in Table 1. The

**Table 1.** Characteristics of the two RES systems. Range resolution and horizontal sampling distance are given for the processed data.

| System | Source wavelet | Center freq. (MHz) | Bandwidth/ pulse length | Range resolution (m) | Horizontal sampl. * (m) | Reference |
|---|---|---|---|---|---|---|
| AWI | burst | 150 | 60 ns | 5 | 75 | Nixdorf et al. (1999) |
| AGAP-South | chirp | 150 | 20 MHz | 7 | 13 | Lohoefener (2006) |

\* after 10-fold stacking

AWI system has been operated in toggle mode for most surveys, where it is alternating between shots with 60 ns and 600 ns burst lengths. The range resolution of each of these bursts in ice is approximately 5 m and 50 m, respectively (Nixdorf et al., 1999). The recorded data are stacked to an average trace distance of 75 m and a vertical sampling interval of 13.33 ns. We use the 60 ns data for IRH interpretation. To increase the contrast of IRHs for the tracing procedure, we differentiate the data, apply a low-pass filter of 150 MHz and an automatic gain-control filter in vertical direction (Steinhage, 2001). The AGAP-South

survey has a gridded layout with line spacings of 5 km and 33 km in east–west and north–south direction, respectively (Bell et al., 2011). The RES system is based on the Multi-Channel Coherent Radar Depth Sounder (MCoRDS), developed by the Center for Remote Sensing of Ice Sheets (CReSIS). Synthetic-aperture migration was applied. We additionally differentiate the data for consistency with the AWI data. The last step before tracing the IRHs is to shift the surface reflection in each trace to time zero. We find this to be the easiest way to ensure consistency in the surface, bed and IRHs at profile crossovers, as the

radar-system delay is unknown for some profiles and thus there are biases in the measured travel times of the reflections. With this flattened ice surface we obtain the depths below the surface but sacrifice the absolute elevations of the traced IRHs.

     All RES profiles are connected via profile cross-overs within two separate regions. We refer to the regions DML for Dronning Maud Land and CEA for Central East Antarctica. DML includes profiles passing the Dome Fuji (DF) and EDML ice cores. Deep ice cores in the CEA region are the EDC and Vostok ice cores. An overview of the RES profiles and deep ice core

locations in East Antarctica is given in Fig. 1.

### 2.2  Internal reflection horizons

We semi-automatically trace the IRHs in the two-way travel-time (TWT) domain, using Halliburton's seismic-processing package Landmark. We choose the most distinct IRHs that can be traced continuously along the profiles, although many more IRHs are visible on local scales. We trace the zero-crossing from positive to negative amplitude of the respective reflections.



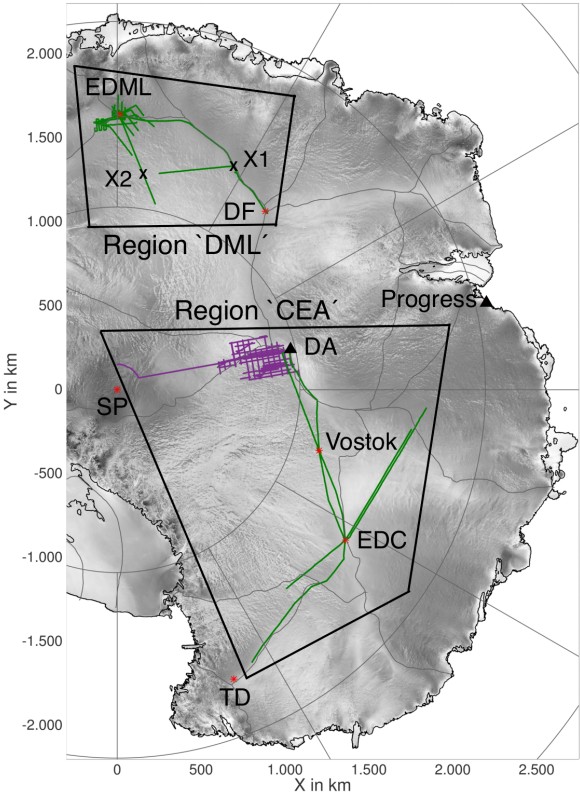

**Figure 1.** Overview of East Antarctica with the RES profiles along which we continuously traced IRHs (green lines: AWI data, purple lines: AGAP-South data) and deep drill sites, stations and other locations mentioned in the text. Abbreviations of sites of interest from top to bottom stand for: EPICA Dronning Maud Land, Dome Fuji, Dome Argus, South Pole, EPICA Dome Concordia and Talos Dome. Gray lines indicate the ice divides (Zwally et al., 2012), coordinates are polar-stereographic.

The TWT of the zero crossing is equivalent to that of the maximum reflection amplitude in the undifferentiated data. The IRHs are transferred from one profile to another at crossover points by comparing the TWTs, which are marked in both profiles by the software. Additionally, we visually compare the reflection patterns in both profiles at the crossover locations. We omit IRHs that cannot be transferred with sufficient certainty, either directly or via wrapping around other lines, or that are truncated or

5   lost in a profile. We do not make any corrections on the IRH depths for the different years of the AWI RES surveys. A different year of data collection causes a slight change in IRH depth. With the maximum time difference between adjacent surveys being only four years and accumulation rates of about 40 to $65\,\mathrm{mm}\,(\mathrm{i.e.})\,\mathrm{a}^{-1}$ in DML, the maximum change in IRH depth is less than 0.3 m of ice. This is less than 5% of the range resolution of the radar system so that we consider it negligible.

    In the DML region we use IRHs that connect the DF and EDML drill sites (Huybrechts et al., 2009; Steinhage et al., 2013).

10   We convert the TWTs of the traced IRHs $t_{\mathrm{IRH}}$ to depths $d_{\mathrm{IRH}}$ (e.g., Steinhage et al., 2001; MacGregor et al., 2015a; Cavitte




et al., 2016):

$$d_{\mathrm{IRH}} = \frac{c_{\mathrm{ice}}\, t_{\mathrm{IRH}}}{2} + z_{\mathrm{f}}, \tag{1}$$

with the constant electromagnetic wave speed $c_{\mathrm{ice}} = 168.5\,\mathrm{m}\,\mu\mathrm{s}^{-1}$ (Gudmandsen, 1975; Peters et al., 2005; Winter et al., 2017) and a constant firn correction $z_{\mathrm{f}}$. In the DML region we use $z_{\mathrm{f}} = 13\,\mathrm{m}$, derived from the measurements of the complex permittivity of five ice cores, down to depths of 100-150 m (Steinhage et al., 2001). We use the relatively recent DFO-2006 (Kawamura et al., 2007) and Antarctic Ice Core Chronology 2012 (AICC2012, EDML) (Bazin et al., 2013; Veres et al., 2013) age scales to assign the IRHs with their ages. The mean age from the DF and EDML age scales is used for the respective IRH. At Vostok, we use the same method of TWT-to-depth conversion, a firn correction of $z_{\mathrm{f}} = 13\,\mathrm{m}$ (Cavitte et al., 2016) and the AICC2012(Vostok) age scale (Bazin et al., 2013; Veres et al., 2013).

Uncertainties of IRH depths are a combination of the vertical accuracy of the radar system and the uncertainty of the TWT-to-depth conversion. The TWT conversion depends critically on the uncertainty of the firn correction (Following Cavitte et al. (2016) we use a variation of $\pm 1.3\,\mathrm{m}$), and the uncertainty of the electromagnetic wave speed in ice. Fujita et al. (2000) give a range from $168.0\,\mathrm{m}\,\mu\mathrm{s}^{-1}$ to $169.5\,\mathrm{m}\,\mu\mathrm{s}^{-1}$ for the wave speed in ice, therefore, we use an uncertainty of $\pm 0.44\%$. This results in an increasing depth uncertainty with increasing IRH depth (e.g., Cavitte et al., 2016; MacGregor et al., 2015a). Even though the accuracy of RES systems for the range estimate for a distinct IRH is always higher than the range resolution of the system we use 5 m radar system uncertainty as an upper boundary. The depth-uncertainty reduction for a distinct IRH depends on the signal-to-noise ratio and is described in the Appendix of Cavitte et al. (2016). The age uncertainty $\Delta a$ of an IRH is the combination of the age uncertainty related to depth uncertainty $\Delta a(\Delta d)$ and the age-scale uncertainty $\Delta a(\mathrm{core})$ at the depth (e.g., MacGregor et al., 2015a):

$$\Delta a = \sqrt{\Delta a(\Delta d)^2 + \Delta a(\mathrm{core})^2}. \tag{2}$$

The TWT-to-depth conversion and age assignment at EDC has been carried out by Winter et al. (2017), who use the ice-core density and conductivity measurements to calculate synthetic radar traces. The synthetic traces are matched against RES data from the vicinity of the ice core site to identify distinct IRHs and their depths (Eisen et al., 2006; Winter et al., 2017). Uncertainties of the IRH depths at EDC depend on the depth uncertainty of the conductivity measurements and the width of the reflection-causing conductivity section and are independent of uncertainties of the wave speed in ice and firn. The depth uncertainties of deep IRHs are thus reduced compared to the TWT-to-depth conversion with Eq. 1. Furthermore, this method assures that the respective IRHs are caused by conductivity contrasts and thus isochronous surfaces (Eisen et al., 2006; Millar, 1981; Siegert et al., 1998b; Fujita et al., 1999) and not of other origin such as changes in crystal orientation fabric (Eisen et al., 2007).

## 3 Results

Starting from EDC, four IRHs were continuously traceable along the central profile in the CEA region, connecting EDC, Vostok and Dome Argus (DA) (Fig. 2 a). The IRHs have ages of $38.2 \pm 0.6\,\mathrm{ka}$, $48.3 \pm 1.2\,\mathrm{ka}$, $90.2 \pm 1.6\,\mathrm{ka}$ and $161.1 \pm 3.5\,\mathrm{ka}$.



**Figure 2.** Sections of some example profiles. The profile locations can be identified via the drill sites and crossing points (vertical lines and labels at the top of sub-figures). Top panels show the elevations (WGS84) of ice surface, bed and IRHs. Colors scale with normalized IRH depths. The second panels show normalized layer thicknesses, with respect to their normalized thicknesses at EDC (a and c), and DF (b and d), respectively.





The 48 ka IRH is not listed in Winter et al. (2017) and is slightly deeper than their horizon H2. The depths of the four IRHs with uncertainties at EDC, Vostok and DA and their ages from the EDC and Vostok age scales with uncertainties are listed in Table 2.

**Table 2.** The depths $d$ below ice surface of the four IRHs and the bed in the CEA region (at Dome C, Vostok and Dome A) with depth uncertainties $\Delta d$ and their ages $a$ from the EDC and Vostok AICC2012 ice-core age scales (Bazin et al., 2013; Veres et al., 2013). The age uncertainties are separated in contributions from depth uncertainties $\Delta a(\Delta d)$ and age-scale uncertainties $\Delta a(\text{core})$.

|      | Dome C | | | | | Vostok | | | | | Dome A | |
| --- | --- | --- | --- | --- | --- | --- | --- | --- | --- | --- | --- | --- |
| IRH  | $d$ | $\Delta d$ | $a$ | $\Delta a(\Delta d)$ | $\Delta a(\text{core})$ | $d$ | $\Delta d$ | $a$ | $\Delta a(\Delta d)$ | $\Delta a(\text{core})$ | $d$ | $\Delta d$ |
|      | (m) | (m) | (ka) | (ka) | (ka) | (m) | (m) | (ka) | (ka) | (ka) | (m) | (m) |
| H1   | 702 | 2 | 38.2 | 0.2 | 0.6 | 572 | 5 | 38.2 | 0.4 | 1.2 | 626 | 5 |
| H2b  | 820 | 6 | 48.3 | 0.6 | 1.0 | 700 | 6 | 48.9 | 0.4 | 1.4 | 758 | 6 |
| H5   | 1269 | 4 | 90.2 | 0.4 | 1.6 | 1284 | 8 | 90.4 | 0.6 | 1.6 | 1102 | 7 |
| H8   | 1892 | 2 | 161.1 | 0.3 | 3.5 | 2147 | 12 | 161.9 | 1.4 | 2.5 | 1425 | 9 |
| bed  | 3241 | 17 | – | – | – | 3743 | 19 | – | – | – | 2076 | 12 |

The shallower IRHs in the profile are relatively smooth because they follow the surface topography. Some locations have shallow IRHs that are steep where the ice crosses very steep bed topography, such as at kilometer 550, shortly before the Vostok ice-core site. Subsequently deeper IRHs gain more features of the bed topography with with lowest IRHs mimicking the broad scale curvature of the bed. The relatively smooth depression in the ice–bedrock interface around kilometer 600 marks Lake Vostok. The downward dip of the stratigraphy over Lake Vostok is clearly visible in the traced IRHs (e.g., Studinger et al., 2003).

The IRH depths along the EDML–DF profile are shown in the top panel of Fig. 2 b. IRH ages from shallow to deep in this profile are: 4.8 ka, 7.6 ka, 10.1 ka, 15.4 ka, 25.0 ka, 38.1 ka, 48.2 ka and 74.2 ka. The oldest IRH is likely related to the Toba eruption (Eisen et al., 2006; Svensson et al., 2013). The IRHs that are most extensively traceable in all other profiles of the DML region are the third deepest and deepest ones with ages of $38.1\pm0.7$ ka, and $74.2\pm1.7$ ka. We focus on these two, most extensive IRHs in the following.

The depths of deepest traced IRHs provide hints about the maximum age of the ice at the respective location. The deeper the IRH with respect to ice thickness, the thinner the ice fraction that is older than the deepest IRH.

The variation of relative layer thicknesses is shown in the bottom panels of Fig. 2a and 2b for the respective profiles. We use layer in the sense of the fraction of ice that is bound by two IRHs, or surface and shallowest IRH. The normalized thickness of each layer is divided by the normalized thickness of this layer at the EDC and DF drill sites, respectively. Especially in the

DF–EDML profile the layer thicknesses systematically and significantly change from DF towards EDML, closer to the coast. The uppermost and lowermost layers show trends of increasing and decreasing layer thicknesses, respectively, from DF to EDML. This layer thickness change is equivalent to deeper IRHs.





The IRH elevations and normalized layer thicknesses relative to EDC for the profile from Talos Dome (TD) towards Progress Station are given in Fig. 2 c. This profile is intersecting the EDC–DA profile at the EDC drill site. The IRHs move deeper and the bottom layer (yellow) becomes thinner with increasing distance from EDC, which means the fraction of ice older than 161 ka becomes smaller. This pattern confirms the results of Frezzotti et al. (2004, 2005, 2007).

Figure 2 d shows IRH depths and layer thicknesses for the profile from DF via crossing point X1 towards X2, concatenated with the approximately perpendicular profile towards EDML via X2. The IRHs become deeper from the plateau toward the coast. The same deepening occurs toward South Pole, as seen in the IRH elevation and normalized depth for the 161 ka IRH in the SP–DA profile (Fig. 2 e). The normalized IRH depth toward SP is in the same range or deeper than at DA, where in turn the normalized thickness of the bottom layer is smaller than at EDC.

The trend of deepening IRHs towards the coast that is visible in the single profiles is confirmed by the spatial distribution of the normalized depths of the IRHs for all RES profiles (Fig. 3 a–3 d). The normalized depths show smooth lateral variations and a broad pattern of deepening IRHs (smaller ice faction older than 161 ka) towards the ice-sheet margins and towards South Pole.

## 4 Discussion

We present a compilation and map the structure of dated IRHs in the EAIS. The depths of the IRHs with respect to ice thickness gives a good first-order indication about the maximum expected age at a location. Assuming a similar spatial pattern of accumulation rates and flow speeds over the last glacial-interglacial cycles, the distance of the 161 ka IRH from the bed, normalized by ice thickness can be considered a proxy of the age at the bottom, compared to the maximum age of undisturbed ice at EDC ($\sim 800$ ka). The normalized depth of the 161 ka IRH at EDC is 0.58. The spatial distribution of normalized depth of 20 the 161 ka IRH is shown in Fig. 3 d. Applying this criterion to our data set, we can tentatively exclude the regions around DA and South Pole (where covered by our data) to be viable for Oldest Ice in the sense of reaching 1.5 Ma old ice at the bottom on a regional scale. A more promising area in this regard could be along the EDC-Vostok profiles shortly before Vostok, where the normalized depth of the 161 ka horizon is smaller than 0.58 and thus its distance to the bed is larger than at EDC (middle panel of Fig. 2 a and Fig. 3 d). However, large features that suggest (buried) megadunes and wind erosion are reported in this region 25 (Cavitte et al., 2016) and also seen in the radargrams of our study. Surface erosion during some time in the past might be the underlying cause for the comparably shallow 161 ka IRH in this region, because the negative surface mass balance facilitates IRHs becoming shallower, instead of deeper over time. The surface erosion causes the disappearance of some layers of ice or snow and disturbs the stratigraphy in the affected depth interval, which impedes the retrieval of an undisturbed record of very old ice.

We find a broad trend of IRHs deepening from the center of the ice sheet towards its margins. This trend is seen in the spatial depth distribution (Fig. 3) and the profiles DF–EDML and TD–Progress Station (Fig. 2 d and 2 c). The most obvious explanation is an increase in accumulation rates towards the coast. However, nonuniform bedrock topography or a changing flow mode can also induce the deepening IRHs (Frezzotti et al., 2005; Leysinger Vieli et al., 2011; Weertman, 1976).



**Figure 3.** The depth distribution of different IRHs, traced in the two regions DML and CEA: (a) 38.2 ka IRH, (b) 74.2 ka IRH, (c) 90.2 ka IRH, and (d) 161.1 ka IRH. The color scale represents IRH depth normalized with ice thickness. Background image indicates ice thickness (Fretwell et al., 2013). Gray lines indicate the position of the ice divides (Zwally et al., 2012).





## 4.1 The accuracy of the IRH mapping

The IRH mapping forms intersecting networks, allowing us to connect the Vostok and EDC deep ice cores with four isochrones. The IRH depths at Vostok and ages from the Vostok age scale can serve as an independent quality control on the tracing procedure because we use only the EDC ice core for the dating of the IRHs. Although the connection of the two ice cores

5 comprises only two RES profiles and misses the gridded survey layout providing crossovers for quality checks (e.g., Cavitte et al., 2016), we find a very good agreement of the IRH ages at EDC and Vostok (Table 2). The differences lie within the uncertainties of the age scales. This gives us confidence that our IRH tracing is reliable despite the unfavorable survey design, provided the IRHs with doubtful pathways are thoughtfully omitted. In comparing our results with Cavitte et al. (2016), we find that our IRH H1 is most probably the same horizon as their youngest isochrone "Reflection 1" because both the depth and

10 age are with in the error bounds at EDC and Vostok. Our lower three IRHs are different ones than the other core-connecting isochrones from Cavitte et al. (2016, Table 3). The age uncertainties of our IRHs are in the same range as in Cavitte et al. (2016), which can be attributed to the dominance of ice core age-scale uncertainty over depth uncertainty contribution.

To further evaluate the quality of our IRH mapping, we conduct a crossover analysis of the IRH depths. Table 3 shows the mean differences of the absolute values and standard deviations of IRH depths for AWI data, AGAP data, and interface of both data types. Cavitte et al. (2016) and Winter et al. (2017) show that the transfer of IRHs between different RES data sets

**Table 3.** Mean values of absolute depth differences $\overline{\Delta d}$, standard deviations $\sigma_{\Delta d}$ and number of cross-overs $N$ from the cross-over analysis for the different IRHs and RES data sets.

| IRH | AWI | | | AGAP-South | | | AWI–AGAP-South | | |
|---|---|---|---|---|---|---|---|---|---|
| | $\overline{\Delta d}$ (m) | $\sigma_{\Delta d}$ (m) | $N$ | $\overline{\Delta d}$ (m) | $\sigma_{\Delta d}$ (m) | $N$ | $\overline{\Delta d}$ (m) | $\sigma_{\Delta d}$ (m) | $N$ |
| 38 ka | 1.6 | 1.6 | 939 | 15.4 | 17.4 | 121 | 5.9 | 4.4 | 18 |
| 74 ka | 2.8 | 5.2 | 939 | | | | | | |
| 161 ka | | | | 15.7 | 16.6 | 180 | 8.9 | 9.1 | 20 |

is possible, and it is so quite smoothly for data with similar range resolution. As the range resolutions of our two data sets are similar (5 m and 7 m), we do not expect major additional uncertainties from the transfer. The calculated crossover errors of the IRH depths are larger within the AGAP-South grid than at the transfer between the AWI and AGAP data. Based on this crossover analysis, we conclude that the transfer of IRHs between the different data sets is reliable within the range resolution.

20 ## 4.2 Factors constraining spatial extent and age resolution of the stratigraphy

We expand the EDC-Vostok stratigraphy (with four IRHs) further to the DA region and into the AGAP-South grid. In the AGAP-South grid, we lose two of the IRHs. The topography of the Gamburtsev Subglacial Mountains causes steep dip reflections that render deep reflections untraceable with the reliability of other regions. Increasing the horizontal sampling resolution





by using the unstacked data (average trace distance of 1.3 m) to avoid destructive stacking due to the phase shift of the reflection from a dipping IRH (e.g., suggested by Holschuh et al., 2014) brings only little improvement. Another possibility to extend the stratigraphy across these regions that could be worth trying is to use unmigrated data, because synthetic-aperture migration also can reduce the return power of steep reflectors (Holschuh et al., 2014). In some regions of the Gamburtsev Mountains,

IRHs are also discontinuous over a portion of the ice column due to recent or former mega dunes and surface erosion by wind scour (e.g. Arcone et al., 2012a, b; Das et al., 2013; Scambos et al., 2012). Cavitte et al. (2016) show that the same structures also exist between DA and Vostok and between Vostok and EDC. These features also hamper the connection of the EDC and Vostok ice cores with additional IRHs in our study.

In DML the IRHs cannot be traced very far from the EDML drill site towards the ice-sheet margin before they are disrupted

or disappear. We attribute this mainly to the onset of the faster ice flow and a changing flow mode (e.g., transition to higher basal sliding or plug flow) towards the ice streams, which causes a disrupted stratigraphy (Bingham et al., 2007; Karlsson et al., 2012; Leysinger Vieli et al., 2011; Rippin et al., 2003).

Another factor constraining the spatial extent of our age stratigraphy is the data availability, such as for the $\sim 900\,\mathrm{km}$ long gap between DF and DA that prevents connection of the DML and CEA regions. When this gap is closed, four deep drill sites

and their age scales will be connected and a zonal section of the age stratigraphy through much of the EAIS exists. We therefore recommend a survey to cross this gap possibly through a joint community effort, such as the AntArchitecture framework.

## 5 Conclusions

We provide an age-depth stratigraphy of two separate regions of the EAIS. In DML the ages of the spatially most extensively traced IRHs are 38.1 ka and 74.2 ka. In the CEA region we trace IRHs with ages of 38.2 ka 48.3 ka, 90.2 ka, and 161.1 ka

and use them to directly connect the EDC and Vostok drill sites and DA. The constraining factor for the spatial extent and age resolution of our stratigraphy is the continuity of the IRHs. In some regions of the Gamburtsev Mountains, traceability of IRHs is hampered by their indiscriminability with the RES system, because of steep slopes. In other parts of the Gamburtsev Mountains and between EDC, Vostok and DA, IRHs in some depth ranges are truncated by features indicating past surface erosion.

The broad picture indicates shallower IRHs on the plateau between DA, Vostok and EDC and deepening IRHs towards the ice-sheet margins and South Pole.

Our data set of traced IRHs with detailed uncertainties provides a valuable complement to other studies that mapped age stratigraphies of the EAIS (e.g., Leysinger Vieli et al., 2011; Cavitte et al., 2016). Establishing the age-depth stratigraphy for the whole EAIS, as aimed for by the AntArchitecture project, is achievable by combined interpretation of data from different

RES systems, the feasibility of which we demonstrated here. We therefore suggest that the next steps towards achieving an age stratigraphy for the EAIS on a continental scale would be

– connecting the mapped IRHs from our study with those from previous studies



- – including already existing data from other RES surveys, such as the most recent extensive Oldest-Ice reconnaissance surveys around EDC and DF, the AGAP-North survey, data collected around South Pole, and the surveys of the International Collaborative Exploration of the Cryosphere through Airborne Profiling (ICECAP) project for a more complete overview

- – closing the gap between DF and DA to connect our two separate regions and facilitate their common interpretation and a direct comparison of age scales of the EDC, Vostok, DF and EDML ice cores.

Joining different data sets for different regions holds the potential to improve the age resolution, either by providing more crossing profiles to circumvent problematic regions, where many horizons are truncated, or complementing strength and weakness of different radar systems.

10 *Data availability.* The IRH depths presented in this study are published on the World Data Center platform PANGAEA: https://doi.pangaea. de/10.1594/PANGAEA.895528

*Author contributions.* **Anna Winter** selected and compiled the data sets, dated the IRHs, transferred IRHs between the data sets, traced IRHs in the AGAP-South data, interpreted the data, created the figures and wrote the manuscript. **Daniel Steinhage** collected the AWI data and is responsible for storage, quality control and IRH tracing in the AWI data. **Timothy T. Creyts** provided the AGAP-South data and 15 strongly supported writing and discussing the manuscript. **Thomas Kleiner** assisted the data interpretation by performing ice-flow modeling studies and contributed to writing and discussing the manuscript. **Olaf Eisen** designed and coordinated the study and contributed to writing and discussing the manuscript.

*Competing interests.* The authors declare that they have no competing interests

*Acknowledgements.* This publication was generated in the frame of Beyond EPICA - Oldest Ice (BE-OI). The project has received funding 20 from the European Union's Horizon 2020 research and innovation programme under grant agreement No. 730258 (BE-OI CSA). It has received funding from the Swiss State Secretariate for Education, Research and Innovation (SERI) under contract number 16.0144. It is furthermore supported by national partners and funding agencies in Belgium, Denmark, France, Germany, Italy, Norway, Sweden, Switzerland, The Netherlands and the United Kingdom. Logistic support is mainly provided by AWI, BAS, ENEA and IPEV. The opinions expressed and arguments employed herein do not necessarily reflect the official views of the European Union funding agency, the Swiss Government or 25 other national funding bodies. AWI RES profiles in the CEA region were supported by PNRA/IPEV, RAE, CHINARE and NIPR. A considerable part of this survey was acquired during the International Polar Year as part of the project TASTE-IDEA. We thank the logistics field team from various nations and flight crews for support during those expeditions. T.T.C. was supported by the US NSF grant PLR-1643970. This is BE-OI publication number xxx.



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
