# Peer review of "Age stratigraphy in the East Antarctic Ice Sheet inferred from radio echo sounding horizons"

_Earth System Science Data, 2018_

## Referee Comment (RC2) · Anonymous Referee #2 · 2 Apr 2019

Summary

This is a concise manuscript on a new hard-earned dataset of general value to the glaciology community, particularly those interested in the long-term history of the East Antarctic Ice Sheet. The strata mapped cross remarkably long distances, perhaps longer than any other comparable effort for certain lines, so in my opinion the manuscript reaches a key threshold of significance in terms of usefulness and completeness. The expected parameters associated with the traced horizons are provided and they placed in appropriate glaciological and geophysical context, and the discussion/conclusions are forward-looking for the value of the dataset. The results themselves are certainly evolutionary rather than revolutionary, but the manuscript is framed appropriately in this regard. Given this context, my concerns are all very minor.

[Figure]

Comments

1/22-24: "Furthermore, an established stratigraphy ensures an undisturbed layering..." appears to be a truism. A better statement of what I think the authors means could be: "Furthermore, mapping of radar-detected horizons increases confidence that layering at those depths is undisturbed..." or something similar.

Figures

Figure 1: Top right corner or map does not display sometimes on screen, which is possibly a graphics export issue? Specify projection used (most likely EPSG:3031). Identify with a different color the sections of transects shown in Figure 2. Add acronyms to caption (only spelled out presently).

Figure 3: Narrow the color scales here to better illustrate the range of IRH normalized depth. Label each panel with isochrone age being shown.

Grammar, etc.

4/7: spell out ice-equivalent 5/21-22: at EDC were carried out...who used the 11/25: merge with previous paragraph

---

## Referee Comment (RC3) · Marie G. P. Cavitte (Referee) · 8 Apr 2019

General comments: This is an important manuscript to publish: it represents the first effort to publish in open access an extensive IRH dataset. The manuscript describes exhaustively how the IRHs were interpreted, dated and assigned uncertainties in both depth and age. The authors do a great job of being exhaustive in all their sources of error, including in the description of sources of error that they then neglect because they are negligible. The authors are also very clear in stating all the issues that affect the spatial extent of their final product which will help future contributors pick up where this manuscript leaves off. I recognize the amount of work involved with publishing a complete IRH dataset on PANGEA, which I applaud. More (myself included) should follow suit.

[Figure]

I went into a little more detail in my review of the manuscript than the other reviewers, but they are all very minor corrections.

Specific comments: I agree with Reviewer 1's comments that some seminal manuscripts on early RES work are missing in the citations and they should be added to the manuscript. I would suggest adding the total age uncertainty in Table 2 even though it is given in the main text. It makes it easier to go back and look up the information. I would also suggest keeping only panel (a) in Fig.3. Indeed, the zoom level is not much different in panels b-d except for the color scale nuances. But panel (a) already shows quite well the pattern of increasing normalized depth toward the coast. However, panel (a) is missing the DA to SP transect present in panel (d), could it be added back to panel (a)? In Table 3, the absolute depth differences are much larger in the AGAP South survey compared to the AWI survey, could the reason why be explained or if unknown hypothesized? Also, I think it would be interesting to have the absolute depth difference in the crossover analysis for the gridded part of the AWI data versus the long single transects (e.g. in CEA). But this might belong more to the realm of future work.

Technical comments: see in supplement

Please also note the supplement to this comment:
https://www.earth-syst-sci-data-discuss.net/essd-2018-140/essd-2018-140-RC3-supplement.pdf

**Supplement:**

Technical comments:
Page 1, L21: add Parrenin et al., 17 as a paper to cite
Page 2, L1: add Parrenin et al., 17 after the Fischer et al., 2013 citation
Page 2, L1: change "largely" to "strongly". And I would actually reword that sentence a little. It currently reads with difficulty
Page 2, L6: change "asses" to "assess"
Page 2, L7: change to "The internal stratigraphy responds to ice sheet dynamics and retains the integrated history of changes in flow..."
Page 2, L8: provide a citation for your statement.
Page 2, L11: change to "...constraints for the evaluation of ice flow..."
Page 2, L21: change to "...has only been obtained over finite areas…."
Page 2, L22: change to "continent-wide"
Page 2, L23: change to "...and the lack of investigations into how..."
Page 2, L26: change "complete" to "whole"
Page 2, L27-28: change to "ice sheet, as demonstrated by MacGregor et al (…) for the Greenland Ice Sheet"
Page 2, L29: first sentence, add some info already about *where* you'll be focussing on… else it feels like a very vague statement
Page 2, L32: remove "with" after "the IRHs are assigned"
Page 2, L33: remove the s from "age scale"
Page 3, L9: add "in **the** vertical"
Page 3, L12: add "processing is" after "migration"
Page 3, L16: change to "...we measure depth below the surface and sacrifice the absolute elevation of..."
Page 3, L17: add "as" after "regions"
Page 5, L5: change "relatively" to "most", or if it is not the most recent, leave off "relatively recent"
Page 5, L7: change to "age scale to assign ages to the IRHs. We use the mean age from the DF and EDML age scales for the respective IRHs."
Page 5, L10-12: change to "Uncertainties **in** xxx"
Page 5, L21: change "has" to "have"
Page 5, L24, 25: change to "Uncertainties **in**" or "IRH depth uncertainties"
Page 7, L6: remove one "with"
Page 7, L11: why are ages here given without uncertainties, compared to the other times the authors provided IRH ages?
Page 7, L13-14: the sentence is awkward. Rephrase?
Page 7, L18-19: The wording is unclear. I don't think until now the concept of "normalized thickness" has been defined. Is it the layer thickness divided by the ice thickness? It would help to clarify. Also, is the normalized layer thickness = layer thickness/their thickness at the cores? Or are we shown the normalized layer thickness / their thickness at the cores? This is unclear to me (Also unclear in Fig.2. caption).
Page 7, L22: The last sentence is too vague.
Page 8, L5: I would use "juxtaposed to" rather than "concatenated with".
Page 8, L7: Saying it is the **same** deepening occurring towards South Pole is confusing, because yes it is deepening at the coast and at South Pole, but the pattern described just a sentence before is one where IRHs become deeper toward the coast. I would say just "A deepening also occurs toward south pole".

Page 8, L15: remove "a compilation" or reword slightly. In my mind, compilation and map are the same thing here?
Page 8, L31: distribution should be plural
Page 8, L34: change to "also induce deepening of the IRHs".
Page 10, L10: I wasn't clear as to what the lowest three IRHs are. Could their names be added here in brackets referring to the Tables?
Page 10, L11: I believe it should be referring to Table 1 from Cavitte et al., 2016?
Page 10, L14: change to "AGAP data, and at the intersections between both data types"
Page 10, L16: change to ", and is relatively easy for radar systems that have a similar.."
Page 10, L22: remove "dip"
Page 11, L3: change to "across these regions would be to use unmigrated"
Page 11, L4: flip "also" and "can"
Page 11, L8: change "with" to "for" and perhaps add that these are not shown in the tables here?
Page 11, L10: Is it "and" or "or" that you meant to use?
Page 11, L15: what is a "zonal" section?
Page 11, L22: Something like "is hampered by the steepness of the IRHs over the steep bed topography" might be clearer.
Page 11, L23: change to "features indicative of past"
Page 11, L25: change to "On large scales, our results indicate shallower IRHs.."
Page 11, L29: change to "is achievable by combining radar data interpretations from.."
Page 11, L30: this is well demonstrated here but also in Cavitte et al., 2016.
Page 12, L7-9: this conclusion could be more clearly worded. Age resolution is a bit vague, I'm assuming it is the age resolution determined by the number of IRHs. This final sentence would benefit being as clear as possible.

Fig.1. The location Ridge B mentioned on Fig.2. should be added to Fig. 1.

---

## Author Comment (AC1) · 2 May 2019

Review 1:
Siegert (Referee)
m.siegert@imperial.ac.uk

I really like this paper. It is about time that East Antarctic internal layers were made available through an open access database; the authors should be congratulated on doing it. I know from first had how challenging this can be, and don't underestimate how much effort has gone into the product. I don't have any major issues with it. There are, however, a few things I might recommend to capture previous relevant work. I will leave it to the authors to respond, if they wish, to these suggestions.

1. The databaseof internal layers from West Antarctica should be mentioned. Siegert, M.J., Pokar, M., Dowdeswell, J.A. & Benham, T. Radio-echo layering in West Antarctica: a spreadsheet database. Earth Surface Processes and Landforms, 30, 1583-1591 (2005).

2. Use of internal layers in calculating past accumulation rates is included in several references, but a these from East and West Antarctica are missing:
Siegert, M.J., Payne, A.J. Past rates of accumulation in central West Antarctica. Geophysical Research Letters, 31, (12), L12403 10.1029/2004GL020290 30 June 2004.
Siegert, M.J. Glacial-interglacial variations in central East Antarctic ice accumulation rates. Quaternary Science Reviews, 22, 741-750 (2003).

3. Use of internal layers in identifying interior ice-flow change. Whillans (1976) is mentioned, but this isn't: Siegert, M.J., Welch, B., Morse, D., Vieli, A., Blankenship, D.D., Joughin, I., King E.C., Leysinger Vieli, G.J.M.C., Payne, A.J., Jacobel, R. Ice flow direction change in interior West Antarctica. Science, 305, 1948-1951. 10.1126/science.1101072 (2004).

4. Exposure of blue ice from internal layers. This paper shows how internal layers can 'outcrop' revealing areas of negative mass balance (sublimation zones) creating blue ice: Siegert, M.J. Hindmarsh, R.C.A. & Hamilton, G.S. Evidence for a large region of surface ablation in central East Antarctica during the last ice age. Quaternary Research, 59, 114-121. (2003).

5. Early work on internal layers, which should be mentioned as we owe much to these early researchers: Clough, J.W. 1977. Radio-echo sounding: reflections from internal layers in ice sheets. Journal of Glaciology, 18, 3-14. Harrison, C.H., 1973. Radio echo sounding of horizontal layers in ice. Journal of Glaciology, 12, 383-397. Millar, D.H.M., 1981. Radio-echo layering in polar ice sheets and past volcanic activity. Nature, 292: 441-443. While some of this uncited work relates to my own, I feel the paper would benefit from including this earlier work in terms of the science behind layers and their usefulness. If the authors would like to include the SPRI layers from EAIS in the database, I'd be pleased to offer them.
Martin Siegert

We thank Martin Siegert for his comments and have included all suggested references in the revised version of our manuscript. We very much appreciate the opportunity to use SPRI layers for future studies, in fact the goal of AntArchitecture. It has been, actually, our intention, that this database will be further extended.

Review 2:
Anonymous Referee #2

Summary
This is a concise manuscript on a new hard-earned dataset of general value to the glaciology community, particularly those interested in the long-term history of the

East Antarctic Ice Sheet. The strata mapped cross remarkably long distances, perhaps longer than any other comparable effort for certain lines, so in my opinion the manuscript reaches a key threshold of significance in terms of usefulness and completeness. The expected parameters associated with the traced horizons are provided and they placed in appropriate glaciological and geophysical context, and the discussion/conclusions are forward-looking for the value of the dataset. The results themselves are certainly evolutionary rather than revolutionary, but the manuscript is framed appropriately in this regard. Given this context, my concerns are all very minor.

Comments
1/22-24: "Furthermore, an established stratigraphy ensures an undisturbed layering. . ." appears to be a truism. A better statement of what I think the authors means could be: "Furthermore, mapping of radar-detected horizons increases confidence that layering at those depths is undisturbed. . ." or something similar.
Figures
Figure 1: Top right corner or map does not display sometimes on screen, which is possibly a graphics export issue? Specify projection used (most likely EPSG:3031). Identify with a different color the sections of transects shown in Figure 2. Add acronyms to caption (only spelled out presently).
Figure 3: Narrow the color scales here to better illustrate the range of IRH normalized depth. Label each panel with isochrone age being shown.
Grammar, etc.
4/7: spell out ice-equivalent 5/21-22: at EDC were carried out. . .who used the 11/25: merge with previous paragraph

We thank the Referee for their comments and included them in the revised version of our manuscript. However, we did not add another color for the sections in Fig. 2, because all the long transects would be marked and just the dense grids around EDML and the Gamburtsevs be left over. It would not easily be possible differentiate the lines of the different radar systems then. And we did not change the color scales in Figure 3. The used color scales are exactly the range of the respective IRH normalized depth, so narrowing  them would have the extremes of the range being lost, I suspect?

Review 3:
Marie G. P. Cavitte (Referee)
mariecavitte@gmail.com

General comments: This is an important manuscript to publish: it represents the first effort to publish in open access an extensive IRH dataset. The manuscript describes exhaustively how the IRHs were interpreted, dated and assigned uncertainties in both depth and age. The authors do a great job of being exhaustive in all their sources of error, including in the description of sources of error that they then neglect because they are negligible. The authors are also very clear in stating all the issues that affect the spatial extent of their final product which will help future contributors pick up where this manuscript leaves off. I recognize the amount of work involved with publishing a complete IRH dataset on PANGEA, which I applaud. More (myself included) should follow suit.

I went into a little more detail in my review of the manuscript than the other reviewers, but they are all very minor corrections.
Specific comments: I agree with Reviewer 1's comments that some seminal manuscripts on early RES work are missing in the citations and they should be added

to the manuscript. I would suggest adding the total age uncertainty in Table 2 even though it is given in the main text. It makes it easier to go back and look up the information. I would also suggest keeping only panel (a) in Fig.3. Indeed, the zoom level is not much different in panels b-d except for the color scale nuances. But panel (a) already shows quite well the pattern of increasing normalized depth toward the coast. However, panel (a) is missing the DA to SP transect present in panel (d), could it be added back to panel (a)? In Table 3, the absolute depth differences are much larger in the AGAP South survey compared to the AWI survey, could the reason why be explained or if unknown hypothesized? Also, I think it would be interesting to have the absolute depth difference in the crossover analysis for the gridded part of the AWI data versus the long single transects (e.g. in CEA). But this might belong more to the realm of future work.
Technical comments: see in supplement

We thank Marie Cavitte for her thorough review and have included most of her suggestions in the revised version of our manuscript. The citations were included, as well as the total age uncertainty in Table 2. However, we would like to show all panels of Fig. 3, as they do not only show a different zoom level, but different isochrones of different ages. Panel (a) does not show the DA to SP transect, because this 38ka isochrone could not be traced along this transect. The 161ka isochrone shown in panel (d), however, was traced along this transect. We clarified the different panels showing different isochrones by including the isochrone ages in the figures.
We thought about the smaller crossover differences in the AWI compared to the AGAP data and guess that they are due to the SAR migration in the AGAP data in combination with the measurements in a region of steep reflectors. The focusing works only along track and thus resolves the depth of dipping reflectors better in the along track direction compared to across track (the footprint is considerably smaller in one direction). Without any focusing (AWI) the depth of the dipping reflector is just the mean depth of the footprint in both along and across track directions. So you get the same depth, no matter in which direction your line is heading compared to the dip of the reflector. But that is just a hypothesis, so we did not include this in the manuscript. The technical comments from the supplement were included.

Best regards
Anna Winter on behalf of all co-authors

---

## Author Comment (AC2) · 2 May 2019

The comment was uploaded in the form of a supplement:
https://www.earth-syst-sci-data-discuss.net/essd-2018-140/essd-2018-140-AC2-supplement.pdf

———————————————————

---

## Author Comment (AC3) · 2 May 2019

attached without track changes in preprint layout. Cari saluti, Olaf

Please also note the supplement to this comment:
https://www.earth-syst-sci-data-discuss.net/essd-2018-140/essd-2018-140-AC3-supplement.pdf